# Effective Elements for Workplace Responses to Critical Incidents and Suicide: A Rapid Review

**DOI:** 10.3390/ijerph18094821

**Published:** 2021-04-30

**Authors:** Tania Pearce, Lyndal Bugeja, Sarah Wayland, Myfanwy Maple

**Affiliations:** 1School of Health, University of New England, Armidale 2351, NSW, Australia; swaylan2@une.edu.au (S.W.); mmaple2@une.edu.au (M.M.); 2Department of Forensic Medicine, Monash University, Southbank 3006, VIC, Australia; lyndal.bugeja@monash.edu; 3Monash Nursing and Midwifery, Monash University, Clayton 3800, VIC, Australia

**Keywords:** critical incident, suicide, workplace interventions, rapid review

## Abstract

Despite high rates of critical incidents (CIs) in working class occupations, there is a significant gap in our understanding of responses to these events. In this study, we aimed to inform a response training module by synthesising the key elements of pre-, during- and post-incident responses to CIs and suicide in the workplace. A rapid review identified studies on responses to CIs or suicide deaths in the workplace published between January 2015 and June 2020. A systematic search of six databases (Medline, CINAHL, PsycINFO, Sociology Collection, Academic Search and Business Search Complete) and grey literature was performed. Studies were excluded if the focus was on non-colleagues. Two reviewers independently conducted record screening, a review of the full text and assessed study quality. The existing evidence was synthesised and interventions were categorised using Haddon’s Matrix. Five studies were included, reporting on CIs across a range of workplace settings, including railways, factories, police and military, along with external critical response units. Overall, study quality was assessed as being poor. Most of the evidence focused on the pre-incident and post-incident stage. There is little evidence on responses to CIs in the workplace. Evidence-based education and training is necessary to establish organisational responses to assist with supporting workers exposed to workplace CIs.

## 1. Introduction

Critical incidents (CIs), including work-related death and injuries, remain a significant public health issue. Recent global estimates report that approximately 350,000 deaths per year are due to fatal incidents, with an additional 313 million workers involved in non-fatal occupational incidents either resulting in serious injuries or requiring at least four days’ absence from work [1]. In the United States of America (USA), during 2018, 5250 total deaths due to work-related injuries were reported, representing a 2% increase from 5147 work related deaths in 2017 [2]. In Australia, 144 workplace deaths were recorded in 2018, with the agriculture and transport industries accounting for the majority [3]. The construction industry also experiences a high number of fatalities, with data from 2018 showing that 24 work-related deaths occurred; a rate of 2.0 per 100,000 employees [3]. Fatalities in the construction industry occur mostly among males with a mean age of 43 years [4]. Despite a downward trend in the number of deaths in the construction industry (30 in 2017–2018), this sector continues to have the highest rate of work-related injury or illness (59 per 1000 employed persons) [5]. In addition to workplace deaths and injuries, another type of CI resulting in occupational trauma is employee suicide. The US Bureau of Labor Statistics (BLS) reported that in 2013, 282 suicides occurred in a work environment (representing six percent of the 4585 total workplace fatalities) [6]. In Australia, a meta-analysis of suicide by occupation found that the risk of suicide was greatest in those industries employing unskilled male workers compared to males occupying highly skilled non-manual positions [7].

The witnessing of injuries (fatal and non-fatal) and suicide or hearing about these types of events from colleagues can leave employees and their employers susceptible to adverse psychological outcomes such as post-traumatic stress disorder (PTSD) [8]. Factors that increase the likelihood of adverse psychological outcomes include the nature of the CI and post-incident events and the amount of exposure and life history of the individual exposed to the event [9]. Symptoms of adverse psychological outcomes include depression, anxiety, insomnia, restlessness and poor concentration [10]. Although only a few individuals will develop chronic mental health issues following exposure to traumatic events [9], workplaces and organisations have become increasingly aware of the need to implement best-practice interventions to help mitigate adverse effects following exposure to occupational trauma.

### 1.1. Evidence-Informed Responses and Postvention for the Workplace

The levels of preparedness to CIs in the workplace vary. Although most workplaces, particularly government organisations, have formal policies and procedures in place, some organisations also choose to outsource their management of critical events. One example involves an early intervention strategy known as employee assistance programmes (EAPs), providing employees with immediate emotional support following traumatic events [11]. An alternative model is the critical incident stress management (CISM) programme [12]. This approach is delivered by mental health professionals and trained workplace peers with the aim of mitigating the effects of exposure to traumatic events [12]. However, a lack of quality studies have resulted in mixed reviews on the effectiveness of CISM. Although anecdotal evidence based on reports of participant satisfaction support the use of CISM [13], scientific advisory councils oppose this type of intervention, citing a lack of convincing evidence of its effectiveness and its potential to cause harm [14,15]. The World Health Organization [16] also rejects the use of CISM, recommending that “psychological debriefing should not be used for people recently exposed to a traumatic event as an intervention to reduce the risk of post-traumatic stress, anxiety or depressive symptoms” (p. 6). Overall, regardless of the approach employed by workplaces when responding to such events, embedded in the policies and procedures of CI strategies are the following goals: “facilitating worker resilience and recovery, reduce subsequent workplace disruption, restore operations and maintain organizational stability” [10] (p. 77).

With regard to suicide postvention—which is defined as: “those activities developed by, with, or for suicide survivors, in order to facilitate recovery after suicide, and to prevent adverse outcomes including suicidal behaviour” [17]—there is little evidence pertaining to supporting people in the workplace [18]. Evidence-based responses to individuals exposed to suicide events are rare; a systematic review of 50 years of postvention research [18] found that only 5% of suicide bereavement research reported interventions. In an update to that systematic review, undertaken in May 2020, only a slight increase to 6.9% was identified. Beyond a bereavement focus, exposure to suicide death has rarely been considered in relation to suicide prevention [19], and there have been no reported evidence-based interventions relating to exposure to death by suicide in the workplace [19].

### 1.2. Training Requirements for Workplaces

There is very little evidence on the type of training workplaces should consider in the management of workplace CIs. Some organisations focus on preventative measures such as pre-incident resiliency training. This approach, which has been implemented in the emergency sector, is focused on developing psychological resiliency and education and awareness among employees in regard to how resiliency can act as a protective factor when exposed to traumatic events [20]. A study on the effectiveness of resiliency training found significant increases in the level of knowledge after post-test assessment compared to pre-test assessment [20]. Another approach is training employees in “psychological first aid” (PFA) which can be delivered either on an individual or group level [21]. PFA is aimed at reducing initial distress and promotion of short- and long-term functioning in those exposed to traumatic events [21]. Evaluations of the effectiveness of PFA are yet to be completed.

To date, due to a focus on the prevention of death by suicide and CIs in the workplace, there has been little attention given to the issues of workplace responses to death by suicide and CIs on an individual level and at the broader organisational level, or of how blue-collar industries are managing responses to these traumatic events.

### 1.3. Aim

The aim of this study was to identify the key elements of responses to CIs and death by suicide in the occupational setting. This was achieved by performing a rapid review of the published peer-reviewed literature on responses to CI and key informant interviews with construction workers and managers about their responses to CIs and deaths by suicide in the workplace. The results of the rapid review only are included here; the broader study findings will be published in the future. Together, these findings informed the development of recommendations for the development of a postvention and CI response training module for Mates in Construction (MiC).

To achieve these goals, this rapid review aimed to:Identify all of the relevant evidence regarding workplace responses to CIs and suicide in the workplace published between 2015 and 2020;Assess the quality of this evidence using the National Health Medical Research Council Hierarchy of Evidence [22] and the Joanna Briggs Institute (JBI) [23] appraisal tools;Synthesise the existing evidence to identify the key elements necessary for the development of an effective workplace programme and;Apply results to a Haddon’s matrix framework to categorise the intervention strategies identified in the review.

## 2. Materials and Methods

### 2.1. Protocol and Registration

This rapid review of evidence was commissioned by Mates in Construction (MiC) Australia to inform the development of critical response training for the period after a suicide or critical incident. Rapid reviews “streamline traditional systematic review methods in order to synthesize evidence within a shortened timeframe” [24]. Studies of rapid reviews suggest this “streamlining” of the review process has minimal impact on the quality of the review produced [25]. Given the short duration of the review, a rapid review approach was selected; however, it was performed and reported in accordance with the Preferred Reporting Items for Systematic Reviews and Meta-Analyses (PRISMA) [26] developed by Moher. A review protocol was not developed.

### 2.2. Information Sources

The following six databases were searched for articles published from January 2015 to June 2020: Medline/Web of Science; Cumulated Index to Nursing and Allied Health (CINAHL)/EBSCO; PsycINFO/ProQuest; Sociology Collection/ProQuest; Academic Search Complete/EBSCO and, Business Search Complete/EBSCO. An example of a search string used to retrieve articles in Medline (Web of Science) is included in Appendix A Table A1. The searches were completed on 15 June 2020 and the results were limited to articles published in peer-reviewed journals. Additional studies were identified by searching grey literature sources, including Open Grey, Trove (National Library of Australia) and Google Scholar, along with the websites of both Australian and International peak organisations (for a list of organisations, see Appendix A Table A2).

### 2.3. Search

Prior to searching the literature, test searches of search strings and databases were conducted. In any quality review, the pre-testing of the search strategy serves a dual purpose. Firstly, testing of keywords and corresponding medical subject headings (MeSH) or synonymous terms (if available) ensures the maximum retrieval of relevant material, while minimising the number of irrelevant records. Secondly, the testing of search strings across a variety of databases guides the process of identifying which database contains information that will help to answer the research question. As a result, eligible studies were identified by searching the titles and abstracts of records using the following search string: (suicide OR death OR critical incident) AND (bereavement OR grief OR mourning OR trauma OR postvention) AND (workplace OR workforce OR employment OR employee OR co-worker OR colleague). Truncation, using asterisk symbol (*) to search variant endings of a word, and/or proximity operators were applied to each of the search terms.

### 2.4. Eligibility Criteria

#### 2.4.1. Inclusion Criteria

Studies were included in the review according to the following a priori criteria: (1) reported on deaths or CIs within the workplace environment; (2) included responses to the incident (e.g., interventions or postvention programmes or strategies); (3) full text available in the English language and published between 2015 and 2020, and, (4) contains original data (qualitative, quantitative, mixed) or a review of original data.

Despite widespread use of the term “critical incident”, no standard definition exists. In the published literature, definitions of the term “critical incident” vary widely depending on the context in which it is being used. A further complication is the frequent use of the term “traumatic event”, which is often used interchangeably with “critical incident”. In a paper on CIs in the police force, Maguen [27] defined a critical incident as: “A potentially traumatic event which may cause a given individual’s emotional resources to become over-taxed, resulting in a spectrum of reactions from exhaustion to increased and unrelenting mental health symptomology” (p. 130). Although Maguen’s definition focuses on the individual, others consider those indirectly affected through sensory or informational exposure, as evidenced in a review paper by Adamson [28] on CIs and best practice in social work. In that paper, a CI is defined as “an event or situation within workplace settings or roles which have the potential to create a sense of emergency, crisis, and extreme stress, or have a traumatic impact on those directly or indirectly affected” (p. 733). A broader interpretation of “critical incident” was generated through the content analysis of fourteen definitions [29]. Fifteen attributes shared amongst the definitions were identified in the results of the study, and a CI was characterised as follows. (1) A CI is a cause of social trauma; (2) it is a cause of fear; (3) it creates an emotional effect on trained people; (4) it causes a change in societal norms; (5) it possibly undermines public trust; (6) it affects the practice of democracy; (7) it is relatively brief in occurrence; (8) it causes significant injury or loss of life, as well as (9) significant property/infrastructure damages; (10) it requires a state of declared emergency; (11) the event is unexpected with (12) a limited scope; (13) it can require intergovernmental/international coordination; (14) it can create positive outcomes and (15) attract significant media coverage [29] (pp. 34, 39–42).

#### 2.4.2. Exclusion Criteria

Studies were excluded if: (1) the focus was on patient or client death (e.g., in a health care or social care setting or on first responders, e.g., police, fire or ambulance workers responding to accidents or suicides that do not involve a work colleague); (2) the study did not contain empirical evidence (e.g., papers that only described an event or intervention); or (3) the paper was a protocol.

### 2.5. Study Selection

Once the searches of the electronic databases and desktop grey literature were completed, the titles and abstracts of the identified records were imported into Endnote x9 bibliographic software. Duplicate citations were removed using Endnote’s duplicate identification tool. A rigorous manual review was also undertaken for any remaining duplicate records. Following this, unique records were imported into Covidence systematic review software [30] for screening and a full-text review. Two reviewers (LB, TP) conducted independent screening of the titles and abstracts of half of the records to determine which of those did not meet the eligibility criteria. Next, the same two reviewers completed screening of the full text of the remaining records. This involved reading each paper in full and determining whether the study met the eligibility criteria. Reconciliation of any conflicts was resolved by a third reviewer (MM).

### 2.6. Data Collection Process and Data Items

Data extraction from the included studies was completed by two reviewers (LB, TP) using a Microsoft Office Excel 2007 (company, city, state, country) coding template, custom made for this project. Categories of data retrieved included country, study design, type, workplace setting, target group, number of subjects, intervention name and description, primary outcome measures and a summary of key results.

### 2.7. Risk of Bias in Individual Studies

Quality appraisal tools determine the reliability of evidence used in systematic reviews by assessing the internal validity of research studies [31]. Evaluation of the internal validity of papers indicates whether the design or methodology of included studies are free from bias [31]. For the purposes of this review, the authors relied on two tools to assess the quality of studies: the National Health and Medical Research Council (NHMRC) Hierarchy of Evidence [22] and the Joanna Briggs Institute (JBI) appraisal tools [23]. For the past two decades, the NHMRC Hierarchy of Evidence has been utilised by evidence-based practitioners to assess the risk of bias of interventional studies [32], whereas the JBI appraisal tools consist of peer-reviewed tools assessing the methodological quality and risk of bias of a range of study types [23]. The methodological quality of included studies was independently assessed by two reviewers (LB and TP) using the JBI appraisal tools. These tools cover 13 study types from cross-sectional and case series through to qualitative studies. A score of ‘1’ was applied for each criterion met and ‘0’ was applied when the criterion was not met or when it was unclear. The number of criteria met were tallied to form the quality score for each study. The scores calculated for each study were then converted to a final quality rating of ‘low’, ‘moderate’ or ‘high’ quality. The following JBI Appraisal Tools and scoring parameters were implemented in this review: Checklist for Analytical Cross-Sectional Studies (score out of 8; low 0–2, moderate 3–5, high 6–8); Checklist for Case Series (score out of 10; low 0–3, moderate 4–6, high 7–10); and Checklist for Qualitative Research (score out of 10; low 0–3, moderate 4–6, high 7–10).

### 2.8. Data Synthesis

The Haddon’s Matrix conceptual framework was applied to capture the key elements of the intervention across time and function. Haddon’s Matrix is a framework that is widely applied in the field of injury prevention and offers a practical approach to recording risk and/or protective factors across time [33]. The columns of the matrix comprise factors relating to the host (i.e., human); agent (i.e., the harm) and environment (both physical and socio-political) and the rows of the matrix comprise the temporal phases of the incident: pre-event, event and post-event, which represent primary, secondary and tertiary responses. Applying Haddon’s Matrix provided a practical approach to capturing information on how to prepare and respond (in the immediate and longer term) to critical events such as workplace death, injury or suicide.

## 3. Results

### 3.1. Study Selection

A total of 1528 studies were located during the peer-review database search, with an additional 69 grey literature records retrieved. After removal of 600 duplicates, 997 records were screened based on title and abstract, resulting in a further 950 records being excluded. Reasons for exclusion at this stage of the screening process included: no empirical data (*n* = 17); no intervention (*n* = 11); not co-worker (*n* = 6); published prior to 2015 (*n* = 4); study examined the impact of CIs on co-workers, not an intervention (*n* = 2); and response to patient or client (*n* = 2). Following the full-text review, five records met the eligibility criteria and were included (see Figure 1 PRISMA flow chart). All included studies were subject to quality assessment.

### 3.2. Study Characteristics

The characteristics of the included studies are presented in Table 1 and Table 2. The five included studies were published between 2015 and 2020 across three countries: The United States of America (*n* = 3), Canada (*n* = 1) and Korea (*n* = 1). Four of the five studies utilised a quantitative methodology and one utilised mixed methods. Three of the four quantitative studies used a cross-sectional design and one was a case series. The one mixed methods study used a quantitative cross-sectional design and a qualitative survey (Table 1). With regard to the types of workplace settings (Table 2), these ranged from railways to factories, police and the military, with one represented by an external critical response unit. In four out of the five examples, interventions were delivered by external professionals who also provided support and advice to both organisations and their employees on managing the psychological effects of traumatic events [10,34,35,36]. Only in one instance was the intervention delivered internally [37].

### 3.3. Risk of Bias within Studies and Level of Study Evidence

As shown in Table 1, among the four studies in which a cross-sectional design was used, three were assessed as “high quality” [10,36,37] and one was assessed as “moderate quality” [35]. The case series study was assessed as “moderate quality” [34], as was the qualitative survey [36]. All studies were classified as NHMRC level IV (i.e., the lowest level of evidence).

### 3.4. Synthesis of Results

#### 3.4.1. Pre-Incident Interventions

Three themes emerged from two studies on pre-incident interventions. Development of worker resilience [36] and development of leadership skills in CI management and recovery were identified as factors relating to the host and organisational preparedness was identified as a factor relating to the socio-political environment (Figure 1) [10,37].

Development of worker resilience centred on enhancing the knowledge of staff and managers about what a CI is and what happens during and after a CI. This included enhancing knowledge on the psychological impact that single and cumulative exposure to CIs can have on workers, in particular emotional reactions such as stress. In addition to impact, studies suggested that staff and managers gain knowledge on strategies they could use to cope with the impact of exposure to CIs, the support networks they could draw on and external specialist resources they could access. Studies also identified education and training interventions for workers in leadership roles on how to manage a CI and adequately support their staff throughout the duration of its impact [37].

*Organisational preparedness* included the design, development and implementation of three key protocols/programmes: CI response and management protocols [37], business and human continuity protocols [10] and peer support programmes [37]. It was also identified that staff and managers gained knowledge of policies on worker entitlements for leave, return to work and legal issues following a CI.

#### 3.4.2. Incident Interventions

Three themes emerged from one study on incident interventions (Table 3)*. Worker support* was identified as a factor relating to the host, incident evaluation was identified was a factor relating to the agent and organisational response was identified as a factor relating to the socio-political environment (Table 2) [37]. Worker support related to compassionate and empathetic communication from managers during and immediately following a CI. Incident evaluation referred to the development of procedures describing whether and how workers should perform emergency responses at the site of a CI, for example, first aid. Organisational response related to the strict adherence to the CI response and management protocols referred to in the pre-incident phase.

#### 3.4.3. Post-Incident Interventions

Six themes emerged from all included studies on post-incident interventions. Intra- and extra-organisational facilitation of worker recovery [34,35,36,37], support for staff in management and leadership roles [10] and facilitation of return to work [36,37] were identified as factors relating to the host. Intra- and extra-organisational and worker recovery were identified as factors relating to the socio-political environment [10,36] (Table 2). Intra-organisational facilitation of worker recovery is related to interventions in place within the organisation for the immediate aftermath of a CI, through to psychological support interventions that could be accessed as required. Studies suggested that immediately following a CI, there should be a requirement that workers are accompanied away from the CI site to a safe location, where they are met by trained peers to offer support as soon as possible [37]. Time off work following the CI was suggested by one study for a period of up to five days, during which regular and compassionate contact is made by the worker’s employer and members of the peers programme [37]. In addition, it was suggested by another study that during this time employers and peers should endeavour to provide validation and understanding of feelings and reactions [36]. For groups of workers that experienced the same CI, psychoeducation [10,35,36] could be offered in addition to an employee assistance programme [37]. 

Extra-organisational facilitation of worker recovery is related to interventions external to but facilitated by the worker’s organisation. All studies identified de-briefing or counselling, with a qualified clinician attending to the worker within three days of the CI. Studies also applied a range of psychological reprocessing techniques, including cognitive behavioural therapy [34,37], eye movement desensitisation and reprocessing [34,36,37], imagery rescripting [34] and erasure and replacement of images [34].

Other extra-organisational interventions included support for staff in management and leadership roles through consultation with an external provider to assist with the restoration of worker and organisational performance [10]. To facilitate the return to work, two studies suggested formal evaluation of the worker’s fitness for work to be performed by a mental health practitioner in collaboration with the worker’s manager [34,36].

As for factors relating to the socio-political environment, intra-organisational and worker recovery interventions were related with a post critical incident seminar presented at six months to two years after the CI [36]. Extra-organisational and worker recovery interventions in one study comprised a monitoring programme for worker and organisational recovery, with the aim of identifying whether additional interventions were required [10].

## 4. Discussion

### 4.1. Summary of Key Findings

To our knowledge, this paper is the first review of interventions designed to respond to CIs and suicide in the workplace. We conducted a systematic search on CIs and suicide in the workplace and evaluated the quality of the evidence. The results of this review demonstrate how scant the evidence base is for individual and organisational interventions to prepare and respond to CIs. Nevertheless, there are some important lessons that can be drawn from the literature. On-site interventions offered to a large number of potentially exposed workers are likely indicated for severe incidents, whereas one-on-one counselling may be sufficient for less severe incidents [10]. Including a variety of personnel from different agencies may be viewed positively, including police and local managers. However, the findings of Bardon [37] indicate that responses from managers can be either positive or negative, with ratings at the scene being more positive, and ratings falling in the days following the incident. When an incident occurs, the rapid removal of workers from the site was important, with by far the majority reporting the average delay of 2 h 41 min as being too long. Sites should expect staff exposed to CIs to take time off work (over 50% took three days off). Intensive and then follow-up work were implemented in workplace disaster settings, as reported by Kang et al. [35], and intensive multi-day sessions were held with a one-month follow up. However, the follow-up was related primarily to outcome evaluation and did not have high participation. Even longer-term interventions may be required to attend to workers for whom the outcomes of exposure to traumatic events continue. Sparn [36] reported on multi-day seminars provided to first responders who had exposure to trauma over longer periods (6 months–2 years) to assist them in working through their trauma stories. This allowed for the normalisation of the experience, as well as the consideration of positive adaptive strategies. This may suggest that immediate, medium and longer-term follow up could be considered. Toukolehto et al. [34] reported on an innovative intervention based on ‘accelerated resolution therapy’ (ART) which aimed to desensitise and reprocess rapid eye movement through mindfulness and re-scripting to erase and replace traumatic images. This intervention, albeit on a small scale, demonstrated success in achieving the stated aims.

#### 4.1.1. Haddon’s Matrix

At which point these interventions may be useful to apply to exposed workers was mapped to Haddon’s matrix along a temporal line of pre-incident, incident and post-incident (Table 1). Moreover, the appropriate location for this intervention was considered in concentric circles radiating out from the worker to the exposure agent, the physical environment and then to policies, procedures and the socio-political environment. Not surprisingly, by far the most evidence was aimed directly at the exposed worker. Almost no focus was on the environmental exposures and physical workplaces. This finding is consistent with the application of Haddon’s matrix in other injury settings [38]. However, this may be an artefact of the search criteria we utilised, as there are clearly occupational health and safety requirements in most workplaces. Interestingly, preparation for CIs was a focus of Bardon [37]; however, this was primarily information provision, rather than specific training about what can be done to reduce the impact of the incident, should one occur in the future.

#### 4.1.2. Organisational and Co-Worker Resources

Of the information that is available on the impact of CIs on co-workers and organisational responses, this primarily consists of popular non-peer reviewed magazine articles featuring descriptions of unevaluated interventions. For instance, the Guided Response, Intervention and Evaluation for Fatalities (G.R.I.E.F.) programme was developed by a social worker to help employees recover from exposure to workplace deaths [39]. A further example is the 2013 publication, *A Manager’s Guide to Suicide Postvention in the Workplace: 10 Action Steps for Dealing with the Aftermath of Suicide* [40] by Carson. Although it is non-peer reviewed, it offers a detailed and practical approach to managing acute, short- and long-term phases in response to a suicide death. Publicly available resources also include mental health webinars on workplace postvention, such as those recently presented by Mortali et al [41] (2020). In this presentation, it was suggested that workplaces should integrate suicide prevention into health and safety plans using nine practices, such as increasing awareness of suicide prevention and cultivating a culture of caring for others in the workplace. Interestingly, the presenters advocated for a “Stratified Suicide Prevention Programme” to include gatekeeper training and a peer network that are trained in recognising and responding to employee suicidal behavior [39]. However, like the previous examples, there is no evidence base by which to assess how effective this model is in addressing suicide prevention or postvention in the workplace. Collectively, the lack of empirical basis for these interventions meant that they were excluded from this review.

Throughout the grey and academic literature, one finding was commonly reported—that the presence of clear messaging and information about the incident is received positively, whereas poor communication increases distress. Gulliver et al. [42], recommended drawing a ‘family tree’ to determine who needs which information and monitoring when they receive information. Such an activity is practical, easy to implement and a collaborative way to assess risk or triage support in the early period following an event. This could be adapted to preventative work and preparatory planning for future incidents.

#### 4.1.3. Employee Trauma

Although workplace deaths are investigated by the appropriate authorities, recommendations about specific interventions to support the workplace are not apparent among the publicly available outcomes from these authorities. For example, in our national search of coroners’ findings, we located the inquest into the death of Colin Arthur Greaves [43]. In this finding, when describing the impact of the employees’ death on his co-workers, the Coroner reported “Further, those two men and others were very affected by the incident with Mr Hepburn not returning to work and Mr Jones resigning from the ERT” [43]. However, the coroner made no recommendations relating to providing support for the co-workers as witnesses to the event. This is not surprising, as the purpose of the coroners’ investigation in most jurisdictions is to determine the identity of the deceased, their medical cause of death and, in some cases, the circumstances relevant to the death. Coroners’ recommendations are an additional discretionary function, the focus of which is on the prevention of future similar deaths, rather than the postvention response. In addition, only a small proportion of coroners’ findings are publicly available.

Current models used to address exposure to trauma in the workplace include critical incident stress management (CISM), otherwise known as critical incident stress debriefing (CISD) or psychological debriefing. There is little evidence of the effectiveness of this approach. Rather, many suggest that this approach is not beneficial and can be harmful, including the US National Institute for Health and Care Excellence (NICE) [44], which recommends that this method be avoided. By contrast, methods that include active post-event monitoring are recommended. The details of what to monitor for are not explicit and should be a priority, with a triage type system to determine who may need more intensive support after an event and who will accommodate or be resilient to the event. Such a triage system, or risk matrix, could be populated to be site- or industry-specific, utilising a matrix to identify when, where, and how to identify vulnerability.

### 4.2. Strengths and Limitations

The major strength of this study was that a broad literature search was performed and a systematic approach was taken to minimise the likelihood that studies that met the eligibility criteria were missed. Two experienced reviewers screened the studies, extracted data and assessed study quality. However, this review was limited by several factors. First, as a rapid review it was time limited and to ensure it is contemporary a date limitation of the past five years was applied. Thus, papers published prior to 2015 were not retrieved. However, our prior review of evidence on postvention, which systematically reviewed all suicide bereavement and postvention evidence over a 50-year period to 2015, provided a solid foundation. Very few papers met our inclusion criteria. This limits the findings of this paper; however, it also demonstrates the very small evidence-base for organisational responses to critical incidents and suicide. Of those papers included, the evidence was generally poor, limited by study design (as per NHMRC evidence ranking) and was all ranked at the lowest level. The quality of evidence reported utilising the JBI appraisal tools [23] was of medium to high quality. Overall, three of the cross-sectional studies were methodologically stronger than the case series or qualitative studies. In the case series, methodological limitations included poor reporting of participant demographics and outcomes. Studies did not routinely include a comparison group, which meant that the results were limited to the presence and levels of satisfaction with interventions and no associations between interventions and outcomes could be assessed. Furthermore, no interventions in the included studies were evaluated for their impact on reductions in adverse outcomes on workers or organisations. Our review did not locate studies that considered transient workers or workers who work across multiple locations and how they experience CIs or suicide. No papers were longitudinal; thus, how people experience workplace exposure to death by suicide and traumatic incidents over time remains unknown. The cumulative impact of multiple exposures also requires consideration. These will be important considerations for training for postvention in the construction industry.

### 4.3. Implications for Service Delivery and Policy Implementation

Despite the small number of studies in this review, there was evidence that some interventions warrant consideration for an organisational approach to CIs. However, it should be noted that few of the studies evaluated these interventions and, as a result, additional examination of the effectiveness of interventions is required. This is specifically needed in order to establish evidence-based education and training for workers, on CI preparedness, and for managers and senior leadership, on support for workers exposed to CIs. The following five recommendations are for consideration and are intended to provide practical guidance in order to develop an organisational approach to prepare for, respond to and recover from CIs.

#### 4.3.1. Recommendation 1: Develop a Critical Incident Preparedness, Response and Recovery Plan

Consistent with a multifactorial approach, interventions could be activated at the relevant temporal phase of CIs—preparedness, response and recovery at both the worker, management and whole-organisation level. As suggested by the authors, these combined interventions could be documented, delivered and monitored through a detailed critical incident preparedness, response and recovery plan (CIPRRP). A CIPRRP could be developed in collaboration with appropriately qualified occupational health and safety practitioners, and could have representation at all levels of the organisation and include business and human continuity protocols. This would ensure that the CIPRRP was based on best-practice principles of occupational health and safety and that there was input from workers, managers and senior leadership in order to maximise investment in the plan. The plan should be managed through a governance structure, be accessible to all staff to ensure transparency and be regularly reviewed (e.g., annual and/or following a CI) and updated.

Measures to evaluate whether a CI was managed in accordance with the CIPRRP should be included in the plan. This could be conducted for both simulated and actual CIs as a mechanism of examining adherence and to identify areas of improvements. Outcomes relating to the evaluation of simulated and actual CIs should be disseminated to all members of the organisation to maintain transparency and as a mechanism of organisational engagement. How, by whom and when such a plan is developed should be included in any training.

#### 4.3.2. Recommendation 2: Establish an Evidence Base for Peer-Support Programmes

Peer support programmes have been implemented in a number of workplaces and were an intervention present in one of the included studies. Further examination of the elements of peer support programmes and their effectiveness in contributing to worker recovery following exposure to a CI may be required to determine whether it should be included in a CIPRRP. Where peer support is used, there is emerging evidence that normalisation of experiences is a useful activity. However, the evidence as to how benefits are monitored and potential challenges are addressed need to be carefully considered.

#### 4.3.3. Recommendation 3: Develop and Deliver Evidence-Based Training on CI Preparedness

CI preparedness could also include all members of organisations receiving practical and evidence-based training on the nature, impact and recovery from CIs. Additional training could be provided to managers and senior leadership on how to support workers exposed to a CI, and that support should be provided at every layer of the organisation. Evidence of participation and passing of an assessment testing workers and managers knowledge could be linked to performance plans to maximise compliance by employees with poor engagement in occupational health and safety.

Consideration could also be given to senior leadership receiving regular external psychological support or at specified intervals following a CI. This will ensure that support is provided by senior leadership to managers that are supporting workers, as well as for senior leadership, as the impact of CIs can be felt beyond directly exposed workers. This training should ideally be undertaken at the time of induction to an organisation and also include regular CI simulations (with post-simulation debriefings) and refreshers.

#### 4.3.4. Recommendation 4: Develop and Deliver Evidence-Based Training on CI Response

The organisational expectations on workers’ and managers’ responses to CIs should be outlined in detail in the CIPRRP and be communicated via an education and training programme, preferably with practical activities and an assessment. Elements that could be considered for inclusion comprise: (1) how workers should respond immediately prior to and during a CI, for example, the extra-and intra-organisational communication of the occurrence of a CI, the provision of first aid to injured co-workers, etc.; (2) extra-and intra-organisational communication by managers and senior leadership; (3) how and what support will be provided to workers in the immediate period following the CI, including, transportation away from CI site (where relevant and possible), including peer and professional psychological support during the period of compassionate leave, along with any human resource and legal considerations.

#### 4.3.5. Recommendation 5: Develop Worker and Organisational Recovery Protocols

Studies identified a number of intra- and extra-organisational interventions to support recovery for workers exposed to CIs and for and organisations impacted by CIs. The CIPRRP could include a suite of clinical and non-clinical psychological support programmes that could be offered to individuals and groups of workers. Organisations should offer access to psychological interventions immediately following exposure to the CI, throughout compassionate leave and for a period following the return to work, as the willingness to engage may change over time. There is good evidence for the effectiveness of psychological reprocessing techniques, particularly for symptoms of post-traumatic stress [45].

At the organisational level, the CIPRRP could include a protocol that for actual CIs, a whole-organisation post-CI seminar or debriefing be delivered to facilitate worker and organisational recovery and promote transparency in the response and recovery process. Protocols could also be included in the CIPRRP that receive input from external occupational health safety professionals, invited (as required) to assess organisational and worker recovery and to provide any recommendations on future interventions required.

## 5. Conclusions

There is a profound lack of available evidence relating to interventions aimed at preventing workplace trauma. This can lead to poor practices that could increase distress in already vulnerable individuals. Monitoring individuals in the post-incident period has received the most attention; however, how and for whom this is done has not been specified. Our analysis of the reviewed studies emphazies that a broader approach to managing critical incidents in occupational settings, which highlights the pre-incident, incident and post-incident temporal pathway, would assist in mapping when and where training should be considered to alleviate adverse outcomes. The focus on the incident as being the primary point for intervention fails to identify the lessons learnt, or training and development opportunities for when incidents occur in the future. Similarly, considering the individual level, as well as the policy and procedural levels, will assist in the integration of training to support workers post-CI or suicide events on-site or among a work crew.

## Figures and Tables

**Figure 1 ijerph-18-04821-f001:**
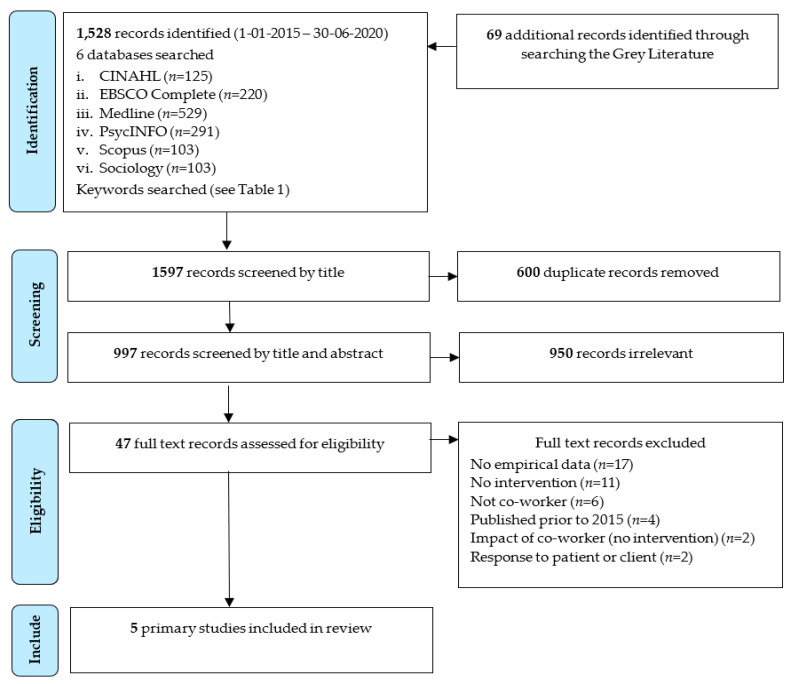
PRISMA flowchart diagram.

**Table 1 ijerph-18-04821-t001:** Study characteristics.

Author (Year)	Country	Methodology	Study Design	NHMRC Level of Evidence	Number of Subjects Participated	Quality Assessment—Score	Quality Assessment—Assessment
DeFraia [10]	United States of America	Quantitative	Cross-sectional	IV	5181	8/8	High
Toukolehto [34]	United States of America	Quantitative	Case series	IV	Not Applicable	5/10	Moderate
Kang [35]	Korea	Quantitative	Cross-sectional	IV	21	5/8	Moderate
Sparn [36]	United States of America	Mixed Methods	Cross-sectional Qualitative Survey	IV	52	7/8 6/10	High Moderate
Bardon [37]	Canada	Quantitative	Cross-sectional	IV	40	7/8	High

**Table 2 ijerph-18-04821-t002:** Study workplace intervention characteristics.

Author (Year)	Workplace Setting	Intervention Name	Intervention Description	Intervention Delivered by	Outcome Measures
Bardon [37]	Railway personnel	Critical Incident Response Programme	On-site intervention and incident management Leaving the site and post-incident employer help Outsourced clinical support Private help seeking	Rail operators	Worker level of satisfaction with Critical Incident Response Programme
DeFraia [10]	Varied (e.g., site managers, medical directors, human resource professionals, union representatives or other organisational officials.)	Critical Incident Response Unit	Distribution of supportive educational materials Interventions to support employees Assistance for managers and leadership Follow-up consultation to ensure ongoing organisational recovery	Staff from an external organisation referred to as occupational health practitioners.	Whether incident severity level influence organisations’ decisions regarding response planning and types of interventions delivered to employees.
Sparn [36]	Police force	Post Critical Incident Seminar	Multiday seminar that provided mental health treatment, peer support and social support	Peers, psychologist and other clinical staff	Post-traumatic stress, depression and anxiety.
Kang [35]	Factory producing textiles	Guidelines for early response to acute stress in the event of a major disaster at a workplace	Disaster response group counselling	Psychologist, industrial hygienist and occupational physician	Impact of event and health
Toukolehto [34]	Military	Accelerated Resolution Therapy-Based Intervention	Mindful awareness and processing of emotions with bilateral eye movements Imaginal exposure and desensitisation Imaginary rescripting of a new positive version for the traumatic event Erasure and replacement of disturbing images Virtual conversations with individuals who were involved in the traumatic event Processing of residual emotions and images	Psychiatrist	Acute stress and grief symptoms

**Table 3 ijerph-18-04821-t003:** Haddon’s matrix of workplace interventions to respond to critical incidents.

	Host (Worker)	Agent (Exposure to Critical Incident)	Environment—Physical (Workplace/Incident Location)	Environment—Socio-Political Workplace Policies and Procedures
**Pre-Incident**	**Develop worker resilience** Information and training for staff and managers on: [37] What happens during and after a CI [37] Stress and its effects [37] Typical emotional reactions and ways to cope with them [37] The cumulative impact of experiencing multiple CIs [37] Support networks [37] Outsourced specialised resources [37] **Develop leadership skills in CI management and recovery** Training managers on how to support staff and manage CIs [37]	-	-	**Organisational preparedness** Design and implementation of CI management protocols that account for reduction of risk factors and promotion of protective factors [37] Planning both business and human continuity [10] Provision of information to staff members on [37] Protocols for time off (including the policy on salary) Return to work policies Legal issues. Development and implementation of a comprehensive peer support programme which includes careful recruitment of peers; regular training updates and follow-up) [37]
**Incident**	**Worker support** Support, compassion and empathetic communication from managers at the CI site [37]	**Incident evaluation** Procedures for evaluation of the capacity of workers to proceed with emergency check at the CI site (e.g., first aid) [37]	-	**Organisational response** trictly implemented CI response and management protocol [37]
**Post-Incident**	**Facilitate worker recovery (internal)** Compulsory demobilisation (removal of staff member from the CI site and return to a safe place) [37] Peer support by trained peers offered rapidly after the CI [37] Staff member taking time off work (24 h–5 days) [37] Regular and compassionate contact from employer and peers programme [37] Validation and normalisation of feelings and experiences, recognition of social support and increased knowledge and understanding of feelings and reactions [36] Psychoeducation in groups who have experienced same type of CI [10,36] Employee assistance programme [37] **Facilitate worker recovery (external)** Clinical de-briefing/meeting with psychiatrist within a few days/96 hours after the CI [34,37] Supportive educational material [10] One-on-one counselling [10,35] Cognitive behavioural therapy (CBT)/processing of residual emotions^,^ [34,37] Eye movement desensitisation and reprocessing (EMDR) to improve PTSD symptoms, social functioning, anxiety and impact of the event [34,36,37] Mindful awareness and bilateral eye movements [34] Imagery rescripting [34] Erasure and replacement of images [34] Virtual conversations [34] **Support for the organisation’s management/leadership team** - Assistance/consultation for managers and leaders in order to restore performance [10] **Facilitate return to work** - Formal fitness to work evaluation of staff member in collaboration with manager and mental health support team [36,37]	-	Short term interventions effective in a resource-limited deployment setting [34]	**Organisational and worker recovery (internal)** Post-critical incident seminar (PCIS) (6 months to 2 years post incident) [36] **Organisational and worker recovery (external)** Monitoring worker and organisational recovery to determine need for additional interventions [10]

## Data Availability

All data generated or analyzed during this study are included in this published article.

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
