# Peer review of "Effective Elements for Workplace Responses to Critical Incidents and Suicide: A Rapid Review"

_ijerph, 2021, doi:10.3390/ijerph18094821_

Round 1

Reviewer 1 Report

Respected Authors,

I do appreciate the opportunity to review the manuscript "Effective elements for workplace responses to critical incidents and suicide: a rapid review." The paper elaborates
on types of responses to critical incidents (CIs) in working class occupations. The scientific community should develop this topic. The authors identified crucial research contributing to developing knowledge in the area. The findings enumerated the significance of identified manuscripts.

Generally, I confirm that the manuscript fulfills all the criteria to be published. 
The introduction section highlights the finding on the issue up to date. Further, we have a clear explanation of "critical incident". The methodology is thorough and in detail explains the search process. The results are precise and described sufficiently. Additionally, there are clear recommendations stated based on achieved results. The language of the manuscript is also clear and concise. Therefore, I would recommend the manuscript publications.
The only minor recommendation is to signpost the discussion to become more reader/user friendly. 

Author Response

Comments: The only minor recommendation is to signpost the discussion to become more reader/user friendly. 

Response: Many thanks. We have amended the discussion section to include 4.12, 4.13 and 4.1.4 – these have been highlighted in the amended manuscript.

We trust that these changes strengthen the paper given its significant contribution to suicide prevention strategies and critical incident response.

Tania Pearce

Reviewer 2 Report

The manuscript ijerph-1177331 is devoted to the actual problem of public health – study of identification the key elements of responses to CIs and suicide in the occupational setting. The reviewed article is interesting for scholars and theme of the article meets the scope of the journal. Work is performed at sufficient scientific level and has good quality; the results of study are professionally interpreted. It can have important application value. The manuscript may be considered for publication after minor revision in International Journal of Environmental Research and Public Health. Prior publication of this manuscript following points needs to be addressed:

  • In general, the work is poorly illustrated. The authors have interesting data that can be presented in the form of diagrams and figures. This approach will definitely improve the scientific perception of the work.
  • It would be good to broaden the discussion section and conclusions in the context of a more detailed presentation of ways to resolve the problem.

My decision is minor revision

Author Response

Dear Reviewer,

Many thanks for the comprehensive review of our manuscript, entitled ‘Effective elements for workplace responses to critical incidents and suicide: a rapid review’. We appreciate the additional time to review the responses. Attached is the updated manuscript with edits, as well as inclusions/corrections highlighted for reviewers to note. We have also included in red the additional references include. We have listed edits, by reviewers below:

Comments:
Point 1: In general, the work is poorly illustrated. The authors have interesting data that can be presented in the form of diagrams and figures. This approach will definitely improve the scientific perception of the work.

Re: Point 1: We note the comments regarding tables, signposting and illustrating the findings and have reordered the ways in which the results are presented – grouping table 1 and 2 within the study characteristics section, and table 3 regarding Haddon’s matrix being explicit to the synthesis of the findings.

Point 2: It would be good to broaden the discussion section and conclusions in the context of a more detailed presentation of ways to resolve the problem.

Re: Point 2: The discussion section has been strengthened and the conclusions amended to ensure the context is clearer. See highlighted section in page 6.  

We trust that these changes strengthen the paper given its significant contribution to suicide prevention strategies and critical incident response.

Tania Pearce
Corresponding author

Reviewer 3 Report

Thanks for your research on this under-represented area. Please see my detailed comments in the annotated file attached. The main areas requiring your attention are the following:

  1. Clearer clarification of your research scope in the introduction and aims, including some restructuring of the content.
  2. Clarifications in the methodology.
  3. Separation of Table 2 into two Tables. Relocation of findings (Tables 1 & 2) in the results section.
  4. More details about the results of the quality appraisal. You mention a few in the strengths/limitations section. You might want to reposition those.
  5. More careful and reflective discussion of the results. The discussion section needs some restructuring.

Author Response

13 April 2021

Dear Reviewer,

Many thanks for the comprehensive review of our manuscript, entitled ‘Effective elements for workplace responses to critical incidents and suicide: a rapid review’. We appreciate the additional time to review the responses. Attached is the updated manuscript with edits, as well as inclusions/corrections highlighted for reviewers to note. We have also included in red the additional references include. We have listed edits, by reviewers below:

Reviewer 3

Comments:

Point 1: Clearer clarification of your research scope in the introduction and aims, including some restructuring of the content.

Re: Point 1: The aim has been highlighted in the introduction, and the reordering of the content, now matches the introduction. Please see page 2.

Point 2: Clarifications in the methodology

Re: Point 2: Noted and highlighted, please see page 4.

Point 3: Separation of Table 2 into two Tables. Relocation of findings (Tables 1 & 2) in the results section.

Re: Point 3: Noted. We have now moved Table 1 and 2 to sit with the exploration of study characteristics in the results section, and then identified the synthesis of results, moving table 3 (Haddon’s matrix) to assist with reader understanding. Please see page 10.

Point 4: More details about the results of the quality appraisal. You mention a few in the strengths/limitations section. You might want to reposition those.

Re: Point 4: In relation to quality assessment, we have added a sentence about the reasoning regarding quality assessment and choice of NHMRC and JBI as an evidence check. Please see Page 6 (highlighted)

Point 5: More careful and reflective discussion of the results. The discussion section needs some restructuring.

Re: Point 5: Please see restructured discussions with clearer signposting and recording of information.

Please note we have also added in references to the reference list – see those highlighted in yellow.

We trust that these changes strengthen the paper given its significant contribution to suicide prevention strategies and critical incident response.

Tania Pearce

Round 2

Reviewer 3 Report

Thanks for the revised paper. However, you did not address my comments in the annotated file of the previous version. I attach it once more for your convenience.

Round 3

Reviewer 3 Report

Thanks for your genuine efforts to address my comments. I have a few more remarks, mainly about comments on the previous version which you did not have access to or not persuasively addressed:

-'Exposure to suicide' could mean that someone has attempted suicide. I still believe you need to adopt another phrase such as 'exposure to suicidal events'. The use of the term you adopted becomes problematic when reading across the article. For example, the phrase '...to support exposure to suicide in the community' could read like interventions should encourage suicide attempts. A better expression of this could be "... to support individuals exposed to suicidal events in the community".

-'Exposure to suicide has rarely been considered in relation to suicide prevention' Still not a clear argument about the connection about those too. Do you mean that persons exposed to suicidal events and develop occupational trauma will be more susceptible to commit suicide?

Comment on previous versions not addressed/convincingly rebutted "Once more, explain your focus on the effects of CIs on individuals (i.e., not the whole envelope CI organisational response) and the groups of the latter you are interested in. Is it about the witnesses of the event, people informed about it, persons who saw photos and video footage during the event investigation?

Actually, the above need to be clarified earlier in the introduction. To this point, you have not explained the individuals and teams possibly affected and who are of interest in this study."

-Comment the authors did not understand (apologies!): The 2nd paragraph of section 2.4.1 presenting the definition(s) of CI matches better the introduction.

-Section 2.7: Still you do not explain at this point which score band you considered as a criterion to include studies after the quality appraisal. Later in the article, it is clear you included studies with high and moderate scores, but for the sake of completion of the methodology, your criterion must be mentioned in this section as well.

Author Response

Thanks for your genuine efforts to address my comments. I have a few more remarks, mainly about comments on the previous version which you did not have access to or not persuasively addressed:

Reviewers Comments -'Exposure to suicide' could mean that someone has attempted suicide. I still believe you need to adopt another phrase such as 'exposure to suicidal events'. The use of the term you adopted becomes problematic when reading across the article. For example, the phrase '...to support exposure to suicide in the community' could read like interventions should encourage suicide attempts. A better expression of this could be "... to support individuals exposed to suicidal events in the community".

Response: Where relevant we have inserted the phrase “death by suicide” or “suicide death” to help clarify the context and meaning. We would argue that the phrase “suicidal events” may be regarded as ambiguous as some papers define “suicidal events” as incorporating everything from attempts to suicidal ideation and death by suicide. While there has been long-standing issues with nomenclature in suicidology, exposure to suicide or suicide exposure utilises the ethos of public health model where we are interested in the effect of an exposure to an event, in this instance suicide death.  

-'Exposure to suicide has rarely been considered in relation to suicide prevention' Still not a clear argument about the connection about those too. Do you mean that persons exposed to suicidal events and develop occupational trauma will be more susceptible to commit suicide?

Response – we have clarified this in specifically in relation to response to suicide in the workplace (p.2). As included previously, we have stated the evidence for those exposed to suicide being demonstrated to be at risk of suicidal thinking and behaviours, and suicide death. We refrain from using the term ‘commit suicide’ as this is stigmatising.

Comment on previous versions not addressed/convincingly rebutted "Once more, explain your focus on the effects of CIs on individuals (i.e., not the whole envelope CI organisational response) and the groups of the latter you are interested in. Is it about the witnesses of the event, people informed about it, persons who saw photos and video footage during the event investigation?

Actually, the above need to be clarified earlier in the introduction. To this point, you have not explained the individuals and teams possibly affected and who are of interest in this study."

Response – we have added in a sentence to clarify the groups we are targeting highlighted in green in the introduction (section 1.2).

-Comment the authors did not understand (apologies!): The 2nd paragraph of section 2.4.1 presenting the definition(s) of CI matches better the introduction.

-Section 2.7: Still you do not explain at this point which score band you considered as a criterion to include studies after the quality appraisal. Later in the article, it is clear you included studies with high and moderate scores, but for the sake of completion of the methodology, your criterion must be mentioned in this section as well.

Response – We apologise for any confusion. However, we do not understand the point you are trying to make as all five studies were included in the paper regardless of their quality. We have included a sentence to further clarify this at the end of section 3.1. As is common practice, the inclusion/exclusion criteria are applied then ALL relevant manuscripts are included in the quality assessment.